# Sex Differences in Time-Domain and Frequency-Domain Heart Rate Variability Measures of Fatigued Drivers

**DOI:** 10.3390/ijerph17228499

**Published:** 2020-11-17

**Authors:** Chao Zeng, Wenjun Wang, Chaoyang Chen, Chaofei Zhang, Bo Cheng

**Affiliations:** 1College of Information Science and Engineering, Henan University of Technology, Zhengzhou 450001, China; zengc_bme@csu.edu.cn; 2State Key Laboratory of Automotive Safety and Energy, Tsinghua University, Beijing 100084, China; zhangf14china@gmail.com (C.Z.); chengbo@tsinghua.edu.cn (B.C.); 3School of Vehicle and Mobility, Tsinghua University, Beijing 100084, China; 4Department of Biomedical Engineering, Wayne State University, Detroit, MI 48201, USA; cchen@wayne.edu

**Keywords:** driving fatigue, ECG, heart rate variability, sex difference

## Abstract

The effects of fatigue on a driver’s autonomic nervous system (ANS) were investigated through heart rate variability (HRV) measures considering the difference of sex. Electrocardiogram (ECG) data from 18 drivers were recorded during a simulator-based driving experiment. Thirteen short-term HRV measures were extracted through time-domain and frequency-domain methods. First, differences in HRV measures related to mental state (alert or fatigued) were analyzed in all subjects. Then, sex-specific changes between alert and fatigued states were investigated. Finally, sex differences between alert and fatigued states were compared. For all subjects, ten measures showed significant differences (Mann-Whitney U test, *p* < 0.01) between different mental states. In male and female drivers, eight and four measures, respectively, showed significant differences between different mental states. Six measures showed significant differences between males and females in an alert state, while ten measures showed significant sex differences in a fatigued state. In conclusion, fatigue impacts drivers’ ANS activity, and this impact differs by sex; more differences exist between male and female drivers’ ANS activity in a fatigued state than in an alert state.

## 1. Introduction

It has been reported that nearly 1.3 million people are killed and 50 million people are injured in road traffic collisions each year [1], and driving fatigue is estimated to be responsible for 20–30% of all road fatalities [2,3,4]. Some studies on driving impairment have considered sleepiness and fatigue as similar mental conditions [5], and the term “fatigue” is often used as an overarching term, which includes sleepiness and mental fatigue [6]. There are many psychophysiological symptoms such as tiredness, lack of energy, difficulties to concentrate, loss of interest, and so on, caused by fatigue that dangerously affect driving [7]. Driving requires mental and physical attention and alertness to be performed effectively [8], and fatigue may affect a driver’s attention and vigilance when controlling a vehicle and may result in a disastrous consequence [9].

Predicting a driver’s mental state by measuring his or her fatigue before or during driving is a method that may be used to reduce traffic accidents [9]. Substantial efforts have been devoted to this topic; employed methods can be divided into three categories [10,11]: vehicle-based methods, behavior-based methods and physiological-based methods. Vehicle-based methods detect the level of a driver’s fatigue usually by steering wheel movement or standard deviation in the lane position. Behavior-based methods determine a driver’s mental state by his or her facial movements through a camera. These two categories of methods become apparent only after the driver starts to sleep with limited time left to prevent an accident. Physiological-based methods, in which physiological signals, such as electrocardiogram (ECG) [8,9,10], electroencephalograph (EEG) [8,9,10] and functional near-infrared spectroscopy (fNIRS) [9], are employed, are regarded as reliable and accurate methods for fatigue detection in the early stages. However, these methods are less acceptable for receiving physiological signals because they involve placing electrodes on the body. With the development of noncontact ECG technology [10], the ECG-based method is expected to detect fatigue in a nonintrusive manner while maintaining the advantages of reliability, accuracy and timely detection.

The autonomic nervous system (ANS) controls heart rate by balancing the sympathetic and parasympathetic nervous systems [12]. Several physiological and pathological changes could exert an impact on the ANS. Heart rate variability (HRV) has emerged as the most valuable noninvasive test to assess ANS function [13], and RR interval time series derived from ECGs are the source information for HRV analyses [14]. For the quantitative evaluation of ANS function, time and frequency domain methods have been widely employed in HRV analyses. Several factors, such as smoking [15], stress [16], and anxiety [17], could alter HRV measures. ANS function is also influenced by driving fatigue and could be expressed by HRV measures. The HRV measures absolute power of the VLF band (*P*_VLF(abs)_) and power of the HF band in normalized units (*P*_HF(nu)_) were reported to be significantly increased and power of the LF band in normalized units (*P*_LF(nu)_) and ratio of the LF band power to the HF band power (*r*_HF/LF_) to be significantly decreased in drivers in a fatigued state compared with those in an alert state by Awais et al. [8,10]. Tran et al. [18] reported that a driver’s absolute power of the LF band (*P*_LF(abs)_), *P*_LF(nu)_ and *r*_LF/HF_ increased significantly and that the mean RR interval, *M*_RR_, and *P*_HF(nu)_ decreased significantly. Abtahi et al. [19] reported that the *M*_RR_, standard deviation of RR interval (*D*_RR_), number of successive RR interval pairs that differ by more than 50 ms (*N*_NN50_), *P*_LF(abs)_, absolute power of the HF band (*P*_HF(abs)_) and absolute power of all three bands (*P*_tot(abs)_) increased significantly. These studies contribute to our understanding of the relationship between driving fatigue and ANS function and clarify the development of driving fatigue detection algorithms. A summary of the experimental designs, HRV measures and HRV measure extraction methods of six prior studies related to driver fatigue is provided in Table 1. Sex differences exist in HRV measures [20,21,22], which should be considered when employing these measures.

For the studies related to driving fatigue and HRV measures, two limitations exist. One limitation is that sex differences have been taken into consideration by few (if any) studies, and the other limitation is that the change tendency of some time and/or frequency HRV measures from the driver’s alert state to fatigued state had contradictory results based on the available reports. For studies on the sex differences in HRV measures, to the best of the authors’ knowledge, all of them were researched with subjects in an alert state, leaving the sex differences in HRV time and frequency measures in a fatigued state unknown. Our experiments were performed in a high-fidelity driving simulator, and fatigue was induced by prolonged monotonous, simulated driving. With being aware of the limitations in this field, the aim of the study was three-fold: First, as many HRV measures as possible were extracted using time domain and frequency domain methods to provide a comprehensive evaluation of the impact of fatigue on the driver’s ANS and to provide references for the development of driving fatigue detection methods; second, the HRV measures of drivers of different sexes in different mental states were compared to provide a basis for developing driver-specific fatigue detection algorithms for specific sexes; and finally, the similarities and differences of HRV measures of males and females in both an alert and a fatigued state were compared to provide a new perspective for understanding the sex differences in the ANS.

## 2. Materials and Methods

### 2.1. Subject

Eleven male and nine female college students (age: 25.95 ± 2.67 years) with legal driver’s licenses were recruited to participate in the experiment. A sum of RMB 150 was provided to the subject as compensation for their participation in the experiment. Two of the male subjects were excluded from the ECG-related research. One was excluded due to poor connectivity of the ECG electrodes during the driving period, and the other was excluded due to fatigue throughout the experiment, mainly caused by extensive studying the night before. Subjects were asked to refrain from consuming caffeine, alcohol or tea and from smoking on the testing day. All subjects provided informed consent for inclusion before participating in the study. The study was conducted in accordance with the Declaration of Helsinki, and the protocol was approved by the Ethics Committee of Tsinghua University (20160006). Details of the methods have been described in our previous studies [26].

### 2.2. Driving Simulator

A driving simulator with six degrees of freedom (see Figure 1) was used in the experiment. A passenger car was mounted on the motion base, and five large projections surrounded the car. The driving simulator provided a realistic driving experience to the drivers. To accelerate the development of the driver’s fatigue symptoms, a simulated, straight highway-driving scenario was selected, and there were no other cars in the simulation other than the virtual experimental car during the driving period.

### 2.3. Experimental Protocol

After the experimental details were introduced, single-channel EEG and ECG electrodes were placed on the subject. The Biopac MP 150 system (Biopac Inc., Goleta, CA, USA) was used for signal acquisition. The EEG and ECG signals were acquired at a sampling frequency of 1 kHz. For the ECG, a modified lead II configuration was adopted. The left leg (LL)-equivalent electrode was placed in the upper-left quadrant, and the right arm (RA)-equivalent electrode was placed in the right infraclavicular fossa. Then, a laboratory assistant guided the subject into the car. A pretest of approximately 10 min was performed to familiarize the subject with the experimental environment, followed by a 60 min period of uninterrupted driving. In addition to the EEG and ECG signals, the driver’s facial expressions were recorded by a camera mounted on the center console during the test. The subjects were required to maintain a speed of approximately 75 km/h in the simulated highway scenario. The tests were performed between 9:30 and 17:30, as these are conventional daily working hours. The ECG signals and videos were analyzed offline, and the EEG signals were left for further research.

### 2.4. Fatigue Assessment

The 60-min videos of the drivers’ facial expressions were each divided into 60 one-minute segments. Then, five well-trained experts scored each one-minute segment a score of 1, 2 or 3 according to the criteria in Table 2.

Since the number of segments with a score of 3 was low, the scores of 3 were rescored as 2. In accordance with the length of the ECG signal recommended for short-term HRV analyses [23], the 60-min videos and the ECG signals were divided into 19 five-minute segments, with 40% overlap at each end with the start/end of the adjoining five-minute segments. In each five-minute segment, the scores of every one-minute segment assessed by each expert were summed over. The minimum and maximum of the summed score in a five-minute segment were 25 and 50, respectively. If the summed score is greater than or equal to 38, according to the simple majority principle, the driver can be considered to be in a fatigued state during the five-minute segment, and if the summed score is less than or equal to 37, the driver can be considered to be in an alert state during the five-minute segment. To reduce misjudgments, we adopted the principle of the effective majority, which stipulated that if the summed score was greater than or equal to 40, then the five-minute segment was defined as a fatigued segment; if the score was less than or equal to 35, then the five-minute segment was defined as an alert segment. Otherwise, the segment was defined as a discordant segment. In this manner, we obtained 88 alert segments and 223 fatigue segments for the following analyses. We also obtained 31 discordant segments, which were excluded from further analysis. Figure 2 compares the summed scores of all subjects in 19 five-minute segments. The figure shows that as time elapsed, all subjects essentially changed from an alert state to a fatigued state.

### 2.5. Extraction of HRV Time and Frequency Measures

We detected R peaks in every segment of the ECG, and abnormal beats were detected and corrected with normal R beats. Then, the occurrence times of the R peaks were denoted by a time series (*t*_0_, *t*_1_…, *t*_i_,…), and the RR interval series was obtained. The RR interval series was unevenly sampled, and the HRV signal was assumed to be a continuous signal derived from the RR interval series via cubic spline interpolation. The HRV time series was obtained by sampling the HRV signal evenly at a given sampling frequency of 4 Hz.

The HRV time domain measures were calculated from the RR interval series. To calculate the HRV frequency domain measures, the smoothness priors approach (SPA) with λ = 500, which corresponds to a high-pass filter with a cutoff frequency of 0.035 Hz, was applied to the RR interval series, and the power spectral density (PSD) was obtained by Welch’s periodogram method [27]. Then, the PSD was divided into three bands: very low frequency (VLF, 0–0.04 Hz), low frequency (LF, 0.04–0.15 Hz) and high frequency (HF, 0.15–0.4 Hz).

The abovementioned processes were completed by employing Kubios 3.0.2 (Kubios Oy, Kuopio, Finland) [27]. All the measures were calculated in the manner recommended by the Task Force of the European Society of Cardiology and the North American Society of Pacing and Electrophysiology [23]. The abbreviations of the measures were redefined (see Table 3) in accordance with the requirements of the Chinese national standards.

### 2.6. Statistical Analysis

SPSS 19 (SPSS Inc., Chicago, IL, USA) was used for the statistical analyses. We calculated the means, standard deviations, medians, 1st quartile (Q1) and 3rd quartile (Q3) to describe the distributions of HRV measures for the alert group and fatigued group and for the male-alert (MA) group, male-fatigued (MF) group, female-alert (FA) group and female-fatigued (FF) group to facilitate comparisons with other studies in this field. The Mann-Whitney U test was used to assess whether the time domain and frequency domain measures significantly differed between (1) the alert group and fatigued group, (2) the MA and MF groups and the FA and FF groups, and (3) the MA and FA groups and the MF and FF groups. According to Fritz et al. [29], a common effect size statistic for the Mann-Whitney U test is *r* using the following formula:(1)$$r=\frac{z}{\sqrt{N}},$$
where *N* is the total number of observations, and when runs the test SPSS report the appropriate *z* value. The difference between the means and the 95% confidence interval of the mean difference are also presented for a difference between means convey effect size information [30].

## 3. Results

Table 4 shows the drivers’ HRV measures during the alert and fatigued states. Being fatigued showed significantly (*p <* 0.01) increases in *M*_RR_, *D*_RR_, *D*_RMS_, *N*_NN50_, *p*_NN50_, *P*_VLF(abs)_, *P*_LF(abs)_, *P*_HF(abs)_ and *P*_tot(abs)_ and decreases in *M*_HR_ compared with being alert.

Table 5 and Table 6 presents the HRV measures in the male-alert (MA), male-fatigued (MF), female-alert (FA) and female-fatigued (FF) groups. The MF group showed significantly (*p <* 0.01) increases in *M*_RR_, *D*_RR_, *D*_RMS_, *p*_NN50_, *P*_VLF(abs)_, *P*_LF(abs)_ and *P*_tot(abs)_ and decreases in *M*_HR_ compared with the MA group. The FF group showed significantly (*p <* 0.01) increases in *D*_RR_, *P*_VLF(abs)_, *P*_LF(abs)_ and *P*_tot(abs)_ compared with the FA group.

The HRV measures were also compared between the two sexes. The FA group showed significantly (*p* < 0.01) lower values of *P*_LF(nu)_, *P*_HF(nu)_ and *r*_LF/HF_ compared with the MA group. The FF group showed significantly (*p* < 0.01) higher values of *M*_HR_ and *P*_HF(nu)_, and lower values of *M*_RR_, *D*_RR_, *D*_RMS_, *P*_VLF(abs)_, *P*_LF(abs)_, *P*_LF(nu)_, *r*_LF/HF_ and *P*_tot(abs)_ compared with the MF group.

## 4. Discussion

Using a driving simulator, this study investigated ANS activity in alert and fatigued driving states through HRV time and frequency domain methods. We focused especially on the sex difference in both mental states. For the comparison with existing studies, assessment of the results with and without considering sex differences led to the following points.

### 4.1. HRV Time Domain and Frequency Domain Measures of Drivers without Considering the Difference of Sex

Our results comparing the HRV time and frequency domain measures of drivers in different mental states are shown in Table 4. Table 7 and Table 8 show comparisons of the results of seven studies (six previous studies and this study) related to HRV time and frequency domain measures of drivers in different mental states.

As shown in Table 4, all six HRV time domain measures were significantly different between the alert and fatigued states. As shown in Table 7, *M*_HR_ and *N*_NN50_ are consistent across all the included studies with respect to the change direction from the alert state to the fatigued state.

Although Tran et al. [18] obtained the opposite result, three studies demonstrated that *M*_RR_ is significantly higher in the fatigued state than in the alert state, which is also supported by a review from Ismail et al. [31]. Thus, the influence of fatigue in terms of *M*_RR_ can be determined. *D*_RR_ is likely to increase significantly from the alert state to the fatigued state based on four of the six studies, and *p*_NN50_ is highly likely to increase significantly because it is related to *M*_RR_ and *N*_NN50_, which definitely increase significantly, although the results of this study and that by Tran et al. [18] do not fully agree. The change direction of *D*_RMS_ is still unclear from Table 7.

As shown in Table 8, an increased change direction in *P*_VLF(abs)_ and *P*_tot(abs)_ is consistent across all the included studies, and regarding the powers of the VLF, LF and HF bands, the absolute values are better than the normalized values for discriminating between mental states.

*D*_RR_ reflects the ebb and flow of all the factors that contribute to HRV [28] and correlates with *P*_tot(abs)_, with *r* > 0.9 [32]. Fatigue can be considered a factor that contributes to HRV since significant increases in *D*_RR_ and *P*_tot(abs)_ were confirmed in drivers’ fatigued state compared with their alert state. Given that *p*_NN50_ and *P*_VLF(abs)_ are significantly increased from the alert state to fatigued state, more parasympathetic and sympathetic activities can be inferred to be present in the fatigued state as *p*_NN50_ is often interpreted as a proxy for cardiac parasympathetic activity and *P*_VLF(abs)_ may mirror sympathetic activity [33].

### 4.2. The HRV Time and Frequency Domain Measures of Drivers Considering the Difference of Sex

To evaluate the sex difference, the differences in the impact on the change in mental states in terms of the HRV measures were compared between males and females. Table 6 shows that four HRV measures exhibited significant differences between the alert state and fatigued state in both males and females. For males, another four HRV measures were significantly different between the two mental states, while for females, no other HRV measures were significantly different between the two mental states.

This situation may imply that ANS activities are more sensitive to the change in mental state for males than for females, as HRV is regarded as the most valuable non-invasive test to assess ANS function [13]. The HRV time and frequency domain measures were adopted by Huang et al. [34] and Patel et al. [35] for the establishment of driving fatigue detectors. It could be reasonable to infer that a better result would be obtained for such detectors if the sex factor was considered.

The sex-related influence on the HRV measures was observed by Ryan et al. [20] early in the 1990s. Recently, a comprehensive study on both age and sex differences in HRV measures was reported by Voss et al. [22]. In their study, an HRV analysis was performed on a 5-min ECG recording obtained in the supine position. According to their study, in the 25- to 34-year-old age group, there were four HRV time and frequency domain measures (*M*_RR_, *P*_LF(abs)_, *P*_LF(nu)_ and *r*_LF/HF)_ that were significantly higher in males than in females, one HRV measure (*P*_HF(nu)_) that was significantly lower in males than in females, and five HRV measures (*D*_RR_, *D*_RMS_, *N*_NN50_, *P*_HF(abs)_ and *P*_tot(abs)_) that showed no significant difference between the two sexes. Our study confirmed these results, except for those for *M*_RR_ and *P*_LF(abs)_, which showed a higher median value in males than in females without statistical significance. The difference in the two studies may be that subjects in the study by Voss et al. [22] were in a supine position without a workload, but they were in a sitting position with a driving workload in our study. It should be noted that all studies, including those by Ryan et al. [20] and Voss et al. [22], to our knowledge, have researched the sex difference of HRV measures of subjects in only the alert state.

Investigating the sex-related influence on HRV measures in a fatigued state could enable further understanding of sex differences. However, little is known in this field. In our study, a comparison of the sex-related influence on the HRV measures between an alert state and a fatigued state is shown in Table 6. The sex differences are larger in the fatigued state than in the alert state: seven of the thirteen HRV time and frequency domain measures that show no significant difference between males and females in the alert state become significantly different in the fatigued state. The relationship between stress and HRV measures was reported by Tharion et al. [36], and stress is associated with fatigue [7,32]. Thus, sex differences of HRV measures in subjects in the stress state are highly expected.

The mechanism of the sex differences in the HRV measures has not been clearly disclosed, and our contribution is a new question proposed to the field of neurophysiology: What is the mechanism underlying the sex differences in healthy subjects’ ANS activity in the fatigued state but not in the alert state? From the application point of view, since the HRV time and frequency domain measures in the state of mental fatigue could better characterize the physiological characteristics of sex, whether these measures can better characterize more physiological and pathological characteristics in the state of mental fatigue is a topic worthy of investigation.

In our opinion, an ideal study related to HRV measures and driving fatigue should (1) involve real vehicles with real-world driving scenarios to ensure a real driving experience, (2) assess fatigue in an offline method to allow careful inspection, and (3) extract HRV measures by established software to increase the reproducibility of the experiment by other investigators. None of the abovementioned studies fulfilled all these requirements, thus leading to some disagreements regarding the results to an extent. One limitation of this study is that the experiments were performed in driving simulators rather than in real vehicles, and some differences may exist between our results and those from real driving conditions. Another limitation is that the number of participants is somewhat scarce and presents similar sociodemographic characteristics (age, vocation, etc.), which can limit the generalization of results. Further research using a large sample size with different sociodemographic characteristics will help to further elucidate the findings of this study. Besides, Our experiment was performed between 9:30 and 17:30; however, since the time of day may affect the perception of fatigue, further experiments performed at the same time of day may yield a more elaborate result regarding the relationship between HRV measures and driving fatigue.

## 5. Conclusions

This study investigated the effects of both mental states and sex factors on drivers’ ANSs by extracting thirteen time and frequency domain HRV measures from ECG signals. The main conclusions are as follows: (1) For drivers of both sexes, there are ten HRV time domain and/or frequency domain measures that are significantly different between the alert state and the fatigued state, implying that the HRV time domain and/or frequency domain measures have the ability to characterize drivers’ mental states; (2) For male drivers, nine HRV time domain and/or frequency domain measures are significantly different between the different mental states, and for females, four measures are significantly different between the different mental states. This finding indicates that the HRV time and frequency domain measures have sex differences in characterizing the mental states of drivers; and (3) There are three HRV time domain and/or frequency domain measures in the alert state that show significant differences between male and female drivers, while ten measures in the fatigued state show significant differences between the two sexes. It can be seen that there are more sex differences in autonomic activities in a fatigued state than in an alert state.

## Figures and Tables

**Figure 1 ijerph-17-08499-f001:**
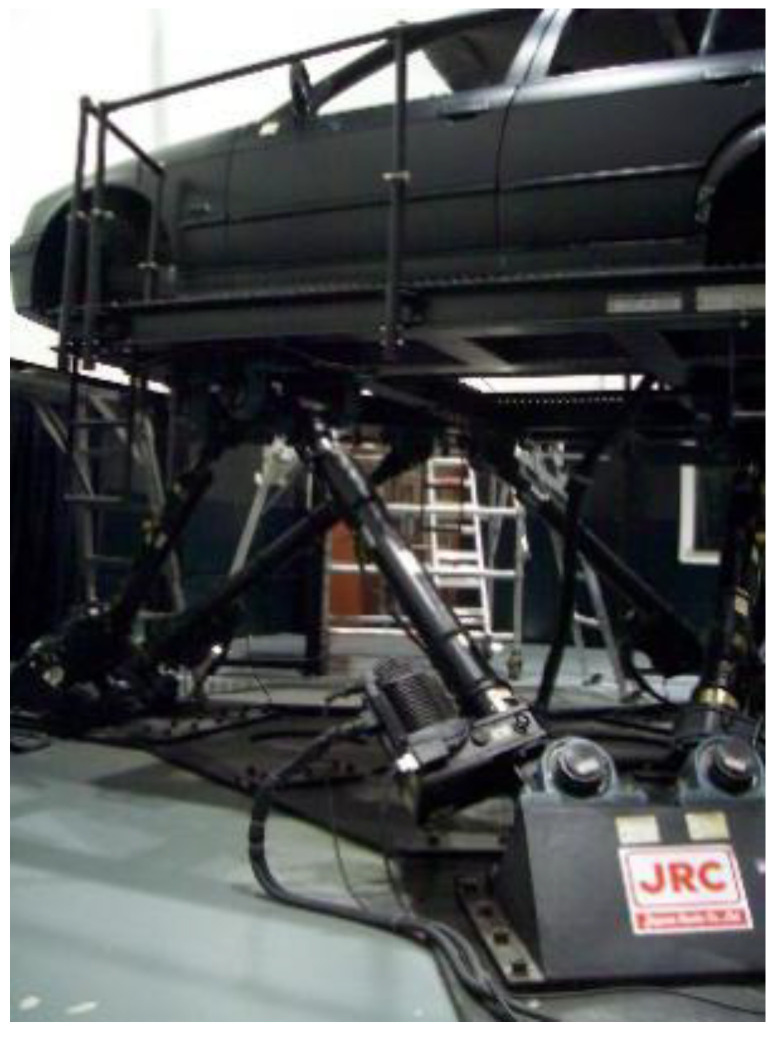
Driving simulator.

**Figure 2 ijerph-17-08499-f002:**
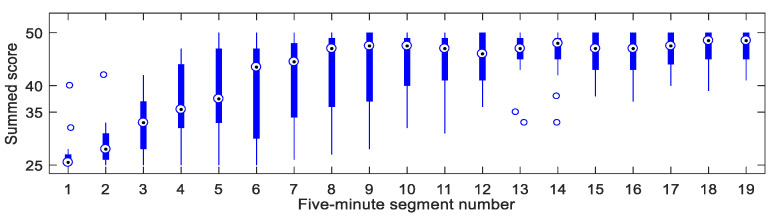
Boxplot of the summed scores of all subjects in 19 five-minute segments.

**Table 1 ijerph-17-08499-t001:** Summary of studies related to HRV measures and driving fatigue.

Number of Subjects (Valid)	Male/Female	Age (Years)	Study Design	Methods Related to Fatigue Assessment	Methods Related to HRV Measure Extraction	References
10 (10)	5/5	29–47	Three real driving sessions on the motorway of approximately 100–105 min each: one during the daytime, one in the evening and one at night	The subjects were asked to provide their Karolinska Sleepiness Scale (KSS) score every 5 min during driving.	HRV analysis was performed following reference [23], and an AR model was used for the PSD estimation of the HRV time series.	[19]
22 (11)	Unknown	18–35	80 min of simulated, monotonous driving	Drowsiness-related events were identified based on a range of facial features, and the driver in the 5 min prior to or after an event was scored as being in an alert or drowsy state.	ECG signal analysis was performed using Biosignal Toolbox.	[8,10]
20 (20)	Unknown	22.6 ± 1.6	120 min of simulated driving with highway scenery	A driver in the 5 min prior to or after driving was scored as being in an alert or fatigued state.	Peak-to-peak intervals were determined through a blood pressure waveform produced by the radial artery. The HRV analysis followed reference [23] and used an FFT for PSD estimation.	[24]
10 (unknown)	Unknown	41 ± 9	Driving on highways for a mean duration of 223 min	Observers classified the state of the driver each minute as either alert or drowsy through video recordings. Ten minutes before and after a drowsy minute was defined as a drowsiness period; the other periods were defined as alert periods.	HRV measures were extracted from a 300-beat window with a step of one beat. The Hodrick-Prescott filter was used for detrending, and a periodogram was estimated using a Hanning window.	[25]
12 (12)	9/3	24–30	Two simulated driving sessions lasting 15–20 min: one in the morning and one in the afternoon after lunch	Two observers scored the drowsiness level according to recorded videos of the drivers’ faces.	HRV measures were extracted from one-minute segments of the ECG.	[5]

**Table 2 ijerph-17-08499-t002:** Subjective scores for driver’s mental state assessment.

Mental State	Score	Description of Facial Expression
Alert	1	The eyes open normally and blink quickly, the eyes are active, the attention appears focused, and attention to the outside world is maintained. The head is upright, and the facial expression is changing frequently.
Fatigued	2	The eyes appear to be partially closed, eyes appear to be partially closed, blinking duration is extended, blinking speed is decreased, eye activity is decreased, or eyes become sluggish; the subject yawns, takes a deep breath, sighs, swallows, rubs the eyelids using their hands, shakes his or her head, scratches his or her face or performs any other action that suggests fatigue or reduced concern with the environment.
Very fatigued	3	The eyes appear to be half or fully closed, the eyelids are so heavy that they are unable to open, the eyes are closed for a long period of time, there is head nodding and head tilting, and the ability to continue driving is lost.

**Table 3 ijerph-17-08499-t003:** Overview of the HRV time and frequency domain measures.

Method	Measure (Abbreviation in This Study)	Measure (Abbreviation in Other Publications)	Description	Unit	Physiological Interpretation
Time domain method	*M* _RR_	Mean RR	Mean RR interval	ms	−
	*D* _RR_	STD RR (SDNN)	Standard deviation of RR interval	ms	Reflects the ebb and flow of all the factors that contribute to heart rate variability. [28]
	*M* _HR_	Mean HR	Mean heart rate	1/min	−
	*D* _RMS_	RMSSD	Square root of the mean squared differences between successive RR intervals	ms	Measurements of short-term variation in the NN cycles and detect high frequency oscillations caused by parasympathetic activity. [13]
	*N* _NN50_	NN50	Number of successive RR interval pairs that differ by more than 50 ms	beats	Reflects parasympathetic activity. [13]
	*p* _NN50_	pNN50	*N*_NN50_ divided by the total number of RR intervals	%	A proxy for cardiac parasympathetic activity. [12,13]
Frequency domain method	*P* _VLF(abs)_	VLF(power)	Absolute power of the VLF band	ms^2^	Increases in resting *P*_VLF(abs)_ power may reflect increased sympathetic activity. [28]
	*P* _LF(abs)_	LF(power)	Absolute power of the LF band	ms^2^	A marker of the parasympathetic tone. [13]
	*P* _HF(abs)_	HF(power)	Absolute power of the HF band	ms^2^	Possibly correlated to sympathetic tone or to autonomic balance. [13]
	*P* _LF(nu)_	LF(pow nu)	Power of the LF band in normalized units	n.u.	A marker of the parasympathetic tone. [13]
	*P* _HF(nu)_	HF(pow nu)	Power of the HF band in normalized units	n.u.	Possibly correlated to sympathetic tone or to autonomic balance. [13]
	*r* _LF/HF_	LF/HF	Ratio of the LF band power to the HF band power	−	An important marker of sympathovagal balance. [13]
	*P* _tot(abs)_	tot(power)	Absolute power of all three bands	ms^2^	The variance of NN intervals over the temporal segment. [13]

**Table 4 ijerph-17-08499-t004:** Descriptive statistics and results of Mann-Whitney U tests of HRV time domain and frequency domain measures of drivers with different mental states.

Measures	Alert (N = 88)					Fatigued (N = 223)					Mann-Whitney U Test	Mean Difference (95% Confidence Interval)	*r*
	Mean	SD	Q1	Med.	Q3	Mean	SD	Q1	Med.	Q3	Alert vs. Fatigued		
*M* _RR_	797.9	137.5	684.3	804.0	870.3	863.5	141.0	738.6	851.4	955.2	*p* < 0.01	65.64 (30.96 to 100.3)	0.20
*D* _RR_	47.06	20.37	33.53	43.05	54.76	65.55	23.87	48.05	60.57	77.67	*p* < 0.01	18.49 (13.17 to 23.81)	0.40
*M* _HR_	77.38	13.0	68.94	74.63	87.68	71.32	11.49	62.82	70.47	81.23	*p* < 0.01	−6.060 (−9.193 to −2.927)	−0.20
*D* _RMS_	33.63	20.44	21.67	27.57	40.18	42.55	23.06	24.56	39.28	45.36	*p* < 0.01	8.915 (3.379 to 14.45)	0.21
*N* _NN50_	46.63	51.36	10.0	19.5	73.75	69.87	55.14	17.0	67.0	89.0	*p* < 0.01	23.24 (9.838 to 36.64)	0.20
*p* _NN50_	14.3	17.8	2.128	5.318	21.76	22.3	20.15	4.239	19.27	26.46	*p* < 0.01	8.002 (3.169 to 12.84)	0.20
*P* _VLF(abs)_	87.25	92.05	32.89	58.0	112.1	201.2	193.1	85.6	136.0	247.5	*p* < 0.01	113.9 (82.00 to 145.9)	0.42
*P* _LF(abs)_	751.8	714.7	297.9	571.4	859.5	1348.0	1056.0	638.5	958.6	1686.0	*p* < 0.01	596.5 (391.8 to 801.3)	0.35
*P* _HF(abs)_	528.6	719.7	159.3	290.5	621.9	701.9	789.5	249.6	470.3	725.4	*p* < 0.01	173.2 (−17.61 to 364.1)	0.17
*P* _LF(nu)_	62.52	18.35	49.65	63.57	77.95	67.78	16.43	57.56	70.49	81.43	*p =* 0.02	5.259 (1.051 to 9.468)	0.13
*P* _HF(nu)_	37.4	18.35	21.95	36.32	50.22	32.09	16.38	18.56	29.43	41.57	*p =* 0.02	−5.304 (−9.505 to −1.103)	−0.13
*r* _LF/HF_	2.667	2.42	0.9887	1.75	3.557	3.155	2.492	1.365	2.395	4.387	*p =* 0.02	0.4887 (−0.124 to 1.101)	0.13
*P* _tot(abs)_	1368.0	1373.0	470.6	949.7	1524.0	2254.0	1730.0	1030.0	1748.0	2911.0	*p* < 0.01	885.2 (517.1 to 1253)	0.34

The mean difference is the fatigue group mean minus the alert group mean.

**Table 5 ijerph-17-08499-t005:** Descriptive statistics of HRV time and frequency domain measures of drivers with different mental states and of different sexes.

Measure	MA	MF	FA	FF
	Mean ± SD, Q1, Med., Q3	Mean ± SD, Q1, Med., Q3	Mean ± SD, Q1, Med., Q3	Mean ± SD, Q1, Med., Q3
*M* _RR_	807.8 ± 155.8, 681.1, 789.1, 962.8	903.8 ± 143.0, 785.7, 927.2, 984.1	787.9 ± 117.4, 684.3, 804.0, 860.4	825.0 ± 128.3, 728.5, 812.4, 858.2
*D* _RR_	50.19 ± 26.94, 30.8, 43.23, 59.21	75.28 ± 28.59, 53.08, 75.8, 92.73	43.93 ± 9.691, 34.6, 42.73, 51.69	56.26 ± 12.66, 46.75, 54.74, 64.15
*M* _HR_	77.01 ± 14.73, 62.33, 76.31, 88.09	68.14 ± 11.39, 60.97, 64.71, 76.55	77.75 ± 11.16, 69.74, 74.63, 87.68	74.36 ± 10.78, 69.91, 73.86, 82.37
*D* _RMS_	34.36 ± 24.38, 16.94, 30.33, 43.77	46.17 ± 25.47, 30.26, 41.42, 50.78	32.91 ± 15.79, 24.19, 27.57, 37.89	39.09 ± 19.99, 24.51, 36.38, 41.64
*N* _NN50_	48.64 ± 54.94, 1.0, 30.0, 78.0	74.97 ± 55.34, 31.5, 69.0, 86.5	44.61 ± 48.07, 13.5, 19.5, 73.0	64.98±54.75, 16.0, 61.0, 89.25
*p* _NN50_	15.57 ± 18.98, 0.2277, 8.506, 23.96	24.79 ± 20.09, 9.427, 21.69, 27.64	13.03 ± 16.66, 3.332, 5.318, 19.53	19.92 ± 20.0, 4.067, 16.93, 24.48
*P* _VLF(abs)_	111.5 ± 118.0, 40.54, 77.98, 146.3	276.3 ± 223.1, 118.0, 217.0, 333.7	63.0 ± 44.98, 32.05, 48.51, 80.22	129.4 ± 122.7, 71.9, 104.4, 152.2
*P* _LF(abs)_	981.8 ± 909.9, 262.7, 681.6, 1371.0	1839.0 ± 1244.0, 876.2, 1547.0, 2506.0	521.8 ± 311.8, 301.9, 516.8, 687.3	878.8 ± 504.4, 516.2, 819.8, 1051.0
*P* _HF(abs)_	617.5 ± 924.6, 77.8, 261.2, 646.3	788.6 ± 968.6, 239.1, 504.9, 746.1	439.7 ± 420.5, 181.8, 290.5, 526.3	618.9 ± 560.1, 251.1, 464.5, 716.5
*P* _LF(nu)_	69.73 ± 14.17, 57.47, 73.59, 79.99	74.24 ± 13.77, 64.08, 79.19, 85.1	55.31 ± 19.33, 38.39, 54.78, 68.06	61.6 ± 16.43, 51.25, 62.04, 75.55
*P* _HF(nu)_	30.2 ± 14.18, 19.98, 26.36, 42.5	25.58 ± 13.63, 14.87, 20.8, 35.81	44.6 ± 19.34, 31.93, 45.1, 61.59	38.32 ± 16.42, 24.38, 37.93, 48.73
*r* _LF/HF_	3.187 ± 2.17, 1.362, 2.792, 4.004	4.176 ± 2.889, 1.788, 3.806, 5.722	2.147 ± 2.567, 0.6233, 1.215, 2.134	2.179 ± 1.503, 1.052, 1.636, 3.1
*P* _tot(abs)_	1712.0 ± 1799.0, 368.2, 1257.0, 2478.0	2908.0 ± 2121.0, 1487.0, 2395.0, 3713.0	1025.0 ± 582.6, 581.2, 925.3, 1201.0	1628.0 ± 879.9, 987.8, 1386.0, 2074.0

Abbreviations: CI, confidence interval; MA, male-alert group; MF, male-fatigue group; FA, female-alert group; FF, female-fatigue group.

**Table 6 ijerph-17-08499-t006:** Results of Mann-Whitney U tests of HRV time and frequency domain measures of drivers with different mental states and of different sexes.

Measure	MA vs. MF			FA vs. FF			MA vs. FA			MF vs. FF		
	Mann-Whitney U Test	Mean Difference (95% CI)	*r*	Mann-Whitney U Test	Mean Difference (95% CI)	*r*	Mann-Whitney U Test	Mean Difference (95% CI)	*r*	Mann-Whitney U Test	Mean Difference (95% CI)	*r*
*M* _RR_	*p* < 0.01	95.97 (33.09 to 158.8)	0.26	*p* = 0.19	37.09 (−25.39 to 99.57)	0.10	*p* = 0.63	−19.93 (−94.98 to 55.13)	0.05	*p* < 0.01	−78.81 (−126.0 to −31.65)	0.32
*D* _RR_	*p* < 0.01	25.08 (15.17 to 34.99)	0.41	*p* < 0.01	12.33 (2.482 to 22.18)	0.43	*p* = 0.81	−6.263 (−18.09 to 5.566)	0.03	*p* < 0.01	−19.02 (−26.45 to −11.58)	0.36
*M* _HR_	*p* < 0.01	−8.870 (−14.26 to −3.484)	−0.26	*p* = 0.19	−3.390 (−8.742 to 1.961)	−0.10	*p* = 0.63	0.7385 (−5.690 to 7.167)	−0.05	*p* < 0.01	6.218 (2.179 to 10.26)	−0.32
*D* _RMS_	*p* < 0.01	11.81 (1.558 to 22.06)	0.25	*p* = 0.05	6.182 (−4.004 to 16.37)	0.15	*p* = 0.73	−1.456 (−13.69 to 10.78)	−0.04	*p* < 0.01	−7.082 (−14.77 to 0.6060)	0.19
*N* _NN50_	*p =* 0.01	26.34 (1.370 to 51.30)	0.22	*p* = 0.04	20.37 (−4.436 to 45.17)	0.16	*p* = 0.23	−4.023 (−33.82 to 25.77)	−0.13	*p* = 0.55	−9.990 (−28.71 to 8.730)	0.04
*p* _NN50_	*p* < 0.01	9.219 (0.2436 to 18.20)	0.23	*p* = 0.05	6.894 (−2.025 to 15.81)	0.16	*p* = 0.42	−2.544 (−13.26 to 8.170)	−0.09	*p* = 0.1	−4.870 (−11.60 to 1.862)	0.11
*P* _VLF(abs)_	*p* < 0.01	164.8 (91.40 to 238.1)	0.47	*p* < 0.01	66.41 (−6.495 to 139.3)	0.41	*p* = 0.02	−48.49 (−136.1 to 39.09)	0.24	*p* < 0.01	−146.86 (−201.9 to −91.82)	0.44
*P* _LF(abs)_	*p* < 0.01	857.7 (453.4 to 1262)	0.37	*p* < 0.01	357.0 (−44.74 to 758.7)	0.38	*p* = 0.04	−460.0 (−942.6 to 22.60)	0.22	*p* < 0.01	−960.7 (−1264 to −657.5)	0.44
*P* _HF(abs)_	*p =* 0.05	171.0 (−183.4 to 525.4)	0.16	*p* = 0.02	179.3 (−172.9 to 531.4)	0.19	*p* = 0.67	−177.9 (−600.9 to 245.1)	−0.05	*p* = 0.64	−169.7 (−435.5 to 96.15)	0.03
*P* _LF(nu)_	*p =* 0.05	4.513 (−2.733 to 11.76)	0.16	*p* = 0.04	6.289 (−0.9107 to 13.49)	0.16	*p* < 0.01	−14.42 (−23.07 to −5.773)	0.38	*p* < 0.01	−12.65 (−18.08 to −7.211)	0.40
*P* _HF(nu)_	*p =* 0.04	−4.620 (−11.84 to 2.604)	−0.16	*p* = 0.04	−6.275 (−13.45 to 0.9033)	−0.16	*p* < 0.01	14.40 (5.778 to 23.02)	−0.38	*p* < 0.01	12.75 (7.327 to 18.16)	−0.40
*r* _LF/HF_	*p =* 0.05	0.9894 (−0.07761 to 2.057)	0.16	*p* = 0.04	0.03274 (−1.028 to 1.093)	0.16	*p* < 0.01	−1.040 (−2.314 to 0.2333)	0.38	*p* < 0.01	−1.997 (−2.798 to −1.197)	0.40
*P* _tot(abs)_	*p* < 0.01	1196 (486.1 to 1906)	0.34	*p* < 0.01	603.0 (−102.4 to 1308)	0.36	*p* = 0.41	−686.3 (−1534 to 161.1)	0.09	*p* < 0.01	−1279 (−1812 to −746.8)	0.34

Abbreviations: CI, confidence interval; MA, male-alert group; MF, male-fatigue group; FA, female-alert group; FF, female-fatigue group. The mean difference is the MF group mean minus the MA group mean for MA vs. MF, the FF group mean minus the FA group mean for FA vs. FF, the FA group mean minus the MA group mean for MA vs. FA and the FF group mean minus the MF group mean for MF vs. FF.

**Table 7 ijerph-17-08499-t007:** Research results related to HRV time domain measures of drivers in different mental states.

Measure	Unit	Alert State (mean ± SD)	Fatigued State (Mean ± SD)	Statistical Method	Level of Significance	Change Tendency	References
*M* _RR_	ms	688.7 ± 84	753.9 ± 103	One-way ANOVA	*	Up	[19]
		889 ± 122	927 ± 132	Paired *t*-test	*	Up	[27]
		865.7 ± 144	845.5 ± 131	Scheffé’s test	*	Down	[18]
		797.86 ± 137.5	863.5 ± 140.98	M-W U test	*	Up	This study
*D* _RR_	ms	40.8 ± 17	53.2 ± 23	One-way ANOVA	*	Up	[19]
		47.7 ± 16.9	58.6 ± 17.3	Paired *t*-test	**	Up	[26]
		63.6 ± 21.1	73.7 ± 24.3	Paired *t*-test	*	Up	[27]
		106.72 ± 30.38	97.07 ± 45.45	One-way ANOVA	NS	None	[5]
		46.7 ± 29	49.4 ± 25	Scheffé’s test	NS	None	[18]
		47.061 ± 20.375	65.555 ± 23.873	M-W U test	*	Up	This study
*M* _HR_	1/min	70.4 ± 8.6	65.6 ± 6.9	Paired *t*-test	**	Down	[26]
		77.382 ± 12.998	71.322 ± 11.488	M-W U test	*	Down	This study
*D* _RMS_	ms	43.2 ± 21.8	43.2 ± 18.9	Paired *t*-test	NS	None	[27]
		28.67 ± 9.40	31.10 ± 22.07	One-way ANOVA	NS	None	[5]
		50.1 ± 41.4	50.2 ± 36.8	Scheffé’s test	NS	None	[18]
		33.635 ± 20.436	42.55 ± 23.058	M-W U test	*	Up	This study
*N* _NN50_	beats	39.0 ± 47	52.8 ± 48	One-way ANOVA	*	Up	[19]
		46.63 ± 51.365	69.87 ± 55.143	M-W U test	*	Up	This study
*p* _NN50_	%	20.1 ± 22.0	18.1 ± 18.0	Scheffé’s test	NS	None	[18]
		14.302 ± 17.799	22.304 ± 20.145	M-W U test	*	Up	This study

Abbreviations: M-W, Mann-Whitney. Levels of significance (for the M-W U test): NS, not significant; * *p* < 0.01. Levels of significance (for other tests): NS, not significant; ** *p* < 0.01; * *p* < 0.05.

**Table 8 ijerph-17-08499-t008:** Research results related to the HRV frequency domain measures of drivers in different mental states.

Measure	Unit	Alert (Mean ± SD)	Fatigued (Mean ± SD)	Statistical Method	Level of Significance	Change Tendency	References
*P* _VLF(abs)_	ms^2^	859.82 ± 114.12	1338.47 ± 121.61	Paired *t*-test	*	Up	[8,10]
		1233.6 ± 773.1	2135.6 ± 1286.7	Paired *t*-test	**	Up	[24]
		87.25 ± 92.054	201.2 ± 193.13	M-W U test	*	Up	This study
*P* _LF(abs)_	ms^2^	222.4 ± 191	449.5 ± 365	One-way ANOVA	*	Up	[19]
		738.3 ± 869.5	825.5 ± 590.3	Paired *t*-test	NS	None	[24]
		1216 ± 686	1789 ± 1248	Paired *t*-test	NS	None	[25]
		511.15 ± 115.47	606.67 ± 162.70	One-way ANOVA	NS	None	[5]
		1179 ± 1520	1581 ± 1792	Scheffé’s test	*	Up	[18]
		751.79 ± 714.69	1348.4 ± 1055.6	M-W U test	*	Up	This study
*P* _HF(abs)_	ms^2^	127.2 ± 121	241.2 ± 212	One-way ANOVA	*	Up	[19]
		506.3 ± 484.2	757.2 ± 538.2	Paired *t*-test	**	Up	[24]
		572 ± 488	576 ± 520	Paired *t*-test	NS	None	[25]
		244.26 ± 101.69	568.33 ± 312.05	One-way ANOVA	**	Up	[5]
		1415 ± 2612	1218 ± 1789	Scheffé’s test	NS	None	[18]
		528.61 ± 719.68	701.85 ± 789.47	M-W U test	*	Up	This study
*P* _LF(nu)_	n.u.	0.54 ± 0.10	0.46 ± 0.08	Paired *t*-test	**	Down	[8,10]
		0.592 ± 0.190	0.515 ± 0.170	Paired *t*-test	NS	None	[24]
		0.501 ± 0.15	0.566 ± 0.15	Scheffé’s test	*	Up	[18]
		0.62518 ± 0.18346	0.67777 ± 0.16430	M-W U test	NS	None	This study
*P* _HF(nu)_	n.u.	0.32 ± 0.08	0.37 ± 0.06	Paired *t*-test	*	Up	[8,10]
		0.406 ± 0.191	0.484 ± 0.170	Paired *t*-test	*	Up	[24]
		0.436 ± 0.16	0.35 ± 0.17	Scheffé’s test	*	Down	[18]
		0.37398 ± 0.18349	0.32094 ± 0.16383	M-W U test	NS	None	This study
*r* _LF/HF_	-	2.1 ± 1.5	2.1 ± 0.9	One-way ANOVA	NS	None	[19]
		2.01 ± 0.98	1.39 ± 0.59	Paired *t*-test	**	Down	[8,10]
		2.0 ± 1.3	1.3 ± 0.9	Paired *t*-test	*	Down	[24]
		3.18 ± 1.58	4.33 ± 2.27	Paired *t*-test	*	Up	[25]
		2.55 ± 1.37	1.01 ± 1.55	One-way ANOVA	*	Down	[5]
		1.5 ± 1.3	2.4 ± 2.3	Scheffé’s test	*	Up	[18]
		2.6668 ± 2.42	3.1555 ± 2.4921	M-W U test	NS	None	This study
*P* _tot(abs)_	ms^2^	373.4 ± 302	741.4 ± 584	One-way ANOVA	*	Up	[19]
		1368.4 ± 1373.3	2253.6 ± 1729.9	M-W U test	*	Up	This study

Abbreviations: M-W, Mann-Whitney. Level of significance (for M-W U test): NS, not significant; * *p* < 0.01. Level of significance (for other tests): NS, not significant; ** *p* < 0.01; * *p* < 0.05.

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
