# Peer review of "Sex Differences in Time-Domain and Frequency-Domain Heart Rate Variability Measures of Fatigued Drivers"

_ijerph, 2020, doi:10.3390/ijerph17228499_

Round 1

Reviewer 1 Report

Authors provided a revision of first version of the manuscript. They responded to most of my comments. Still, some issues are uncovered or misunderstood.

  1. Explanation, why such research is needed and beneficial in the field, is lacking in the text (Introduction part). 
  2. In introduction authors stated that there is inconsistency in the results related to alert and fatigued state, but they did not provide evidence of that. Table 1 has to contain results of the studies, not only methodology.
  3. I was suggesting to describe previous knowledge in Introduction. Authors still have a lot of previous studies references in Discussion. Lines 255-266; 274-282; 283-289 have to be moved and integrated into Introduction.
  4. I still insist that the most important shortcoming of data description is naming only the titles of variables of HRV without explaining their meaning and functional differences. It is impossible to understand what are the differences between genders or states of drivers. In the conclusions only number of differences is stated without understanding the meaning of differences. No explanation/interpretation of those differences is provided in Discussion. Answer to reviewer does not provide the explanation either.
  5. The effect size could be calculated even with nonparametric data (e.g. using G*Power).
  6. 2nd paragraph of conclusions has to be moved to discussion section.

Reviewer 2 Report

I have reviewed both the text extensions added by the authors. I am pleased with the result. Therefore, now I would recommend the publication of this manuscript.

Author Response

Thank you for the remarks.

Reviewer 3 Report

To briefly address the style and writing of this manuscript, I did not find any particular issue that could justify its rejection solely based on the few mistakes or typos that can be found here and there in the text. I think the article is written well enough.

Now, moving on to the contents.

The abstract must be a summary of the whole manuscript. In this case, some brief elements from the introduction and conclusion are missing.

The introduction is adequate, even though some elements on fatigue could be developed more in depth, especially for what concerns their symptomatology and the way they directly affect driving. For instance, using articles such as the following one: Alonso, F., Esteban, C., Useche, S., & López de Cózar, E. (2016). Prevalence of Physical and Mental Fatigue Symptoms on Spanish Drivers and Its Incidence on Driving Safety. Advances in Psychology and Neuroscience, 1(2), 10-18. doi: 10.11648/j.apn.20160102.12. Moreover, the last paragraph includes some elements from the performed experiment that should be moved to the methodology section, leaving the introduction for the objectives of the study.

For what concerns the methods, the number of participants is somewhat scarce and presents similar sociodemographic characteristics (age, etc.), which can limit the generalization of results, and this must be made clear in the limitations of the study.

As for the rest, the methodology is adequate and it is explained in detail both in what concerns the process that was carried out for the experiment and data gathering, and the data analysis.

Regarding the results, it would be convenient that the tables you mention here were placed as close as possible to the specific section they belong to. I understand the difficulty, considering the size of the tables, but it would be more adequate (and easier) for the reader if they were moved right before the discussion. Moreover, the data that are already included in the tables do not need to be repeated in the text.

The discussion is well-explained, contrasting the results with other similar studies.

Finally, in the conclusions, you could briefly explain the contributions of this study to the scientific community and/or society, since it can truly help prevent fatigue at the wheel.

Author Response

This manuscript is a resubmission of an earlier submission. The following is a list of the peer review reports and author responses from that submission.

Round 1

Reviewer 1 Report

A straightforward simulation-based study where the authors explored how different sex has varying impacts on fatigued drivers.  This study has the following weaknesses:

  1. On page 2, a few reasons are provided how this study is contributing to the current body of work.  However, it is not that convincing.  At the very least, the authors need to provide a table comparing their studies with the existing studies.  Also, it would always strengthen to list the specific contributions of any article.
  2. The sample size is quite small.  This is a big weakness of the results presented.  Furthermore, the variability of the age of the participants is low.
  3. In the simulation, only straight highway and singular vehicle driving are considered.  What is the reason behind this?
  4. On page 4, it is mentioned that five well-trained experts scored the driving videos.  How are these experts chosen?  How do the authors define "experts" in this case?
  5. On page 3, the authors referred to their other study for ethical review details.  This must be presented in this study as well.
  6. The writing needs significant improvements.  For instance, on page 2, a number of parameters are presented without introducing them first.
  7. Table 5 could easily be presented in the literature review section.

Author Response

Response to Reviewer 1 Comments

Point 1: 

A straightforward simulation-based study where the authors explored how different sex has varying impacts on fatigued drivers.  This study has the following weaknesses:

1.On page 2, a few reasons are provided how this study is contributing to the current body of work.  However, it is not that convincing.  At the very least, the authors need to provide a table comparing their studies with the existing studies.  Also, it would always strengthen to list the specific contributions of any article.

Response 1:

Thank you for the remarks and suggestions.

Table 5 has been moved to the Introduction, and this table became Table 1 in the modified manuscript.

A literature review could be conducted to address two issues: 1) Studies related to driving fatigue and HRV measures; and 2) studies on the sex differences in HRV measures.

For the first issue, ten papers (manuscript reference [7, 9, 11-18]) were investigated. Two limitations have been disclosed from the literature review on this issue: The first limitation is that sex differences have been taken into consideration by few (if any) studies, and the second limitation is that the change tendency of some time-domain and/or frequency-domain HRV measures from drivers’ alert state to fatigued state was proven to have contradictory results based on the available reports.

Regarding the second issue, three papers (manuscript references [19-21]) were investigated. There are relatively fewer papers that address this issue. One limitation has been disclosed from the literature review on this issue: Reports on this issue were researched with subjects in an alert state, leaving the sex differences in HRV time and frequency measures in a fatigued state unknown.

Finally, we derived the aims of the manuscript based on the above-mentioned literature in the last paragraph of the Introduction.

Point 2: 

2.The sample size is quite small.  This is a big weakness of the results presented. Furthermore, the variability of the age of the participants is low.

Response 2:

For studies in the field of driving fatigue, the number of subjects participates in the experiment is usually 10-20 (manuscript reference [5, 7, 9, 18, 23, 24]). The sample size of this study is 18, which is basically the same as the sample size of the previous studies.

A relatively small age range could ensure that the results of the manuscript are not affected by age factors.

Point 3: 

3.In the simulation, only straight highway and singular vehicle driving are considered.  What is the reason behind this?

Response 3:

According to our experience, the straight highway-driving scenario can induce driving fatigue most effectively.

Point 4: 

4.On page 4, it is mentioned that five well-trained experts scored the driving videos.  How are these experts chosen?  How do the authors define "experts" in this case?.

Response 4:

The research team of the manuscript is led by Professor Bo Cheng from Tsinghua University. The team has been engaged in driving fatigue research for more than ten years (https://www.mendeley.com/authors/36611317200/). The fatigue assessment experts are all members of this team and have rich experience in fatigue assessment.

Point 5: 

5.On page 3, the authors referred to their other study for ethical review details.  This must be presented in this study as well.

Response 5:

Text “All the subjects provided informed consent before the experiment, and all the procedures were approved by the university ethics committee. Details of the methods have been described in our previous studies [25]” just referred to our other study for details of the ECG collection methods rather than for ethical review details. Our ethics approval document could be provided upon request.

Point 6: 

6.The writing needs significant improvements.  For instance, on page 2, a number of parameters are presented without introducing them first.

Response 6:

The text has been modified with a description added before the symbols (measure abbreviations) used for the first time in the Introduction. 

Point 7: 

7.Table 5 could easily be presented in the literature review section.

Response 7:

Table 5 has been moved to the Introduction, and this table became Table 1 in the revised manuscript.

Reviewer 2 Report

Dear authors,

I believe the manuscript have a potential to be acknowledged by readers. It touches barely known topic which might be either an argument for the novelty or random data (artefacts). Therefore, the extensive explanation why gender differences of HRV during alert and fatigued state in driving are necessary to be studied. In other words, the problem that is solved by manuscript and rationale of empirical study must be provided.

The manuscript itself has unusual structure. Previous results aren't described in Introduction part (it might be reasonable to move Table 5 into Introduction). I suggest to provide previous knowledge extensively as literature review. It is important for understanding the topic and impact of current study. Results of current experiment are just mentioned, but aren't described. I deeply believe that description and explanation of obtained results is necessary. 

The most important shortcoming of data description is naming only the variables of HRV. It is impossible to understand what are the differences between genders or states of drivers. It is stated now in the conclusions that groups differed according some number of variables. What is the worth of such conclusions? Authors must extensively describe the differences and explain what is the meaning of those differences. Some neurophysiological mechanisms of fatigue x gender should be discussed.

Discussion of the results is poor. There is only one attempt to interpret some results (line 227), still the reference to it must be provided.

Minor points:

Where any incentives proposed for participation in the study?

Table 3 is not necessary, when all results are presented again separately for men and women.

Table 4 is hardly readable. Authors must to look for the format to present it clearly. I would suggest to split it to two tables (or even three: one for males, next for females, and last for significance between genders).

The effect size for gender or state differences would be beneficial for understanding the data.

It is very untypical to name p<.01 as low significance and p<.001 as high significance. The effect size could be a more reasonable measure of significance in results. 

Only gender differences, but not gender influence could be analysed due to non-random manipulation of the gender (line 176). 

Please, check English of the sentence in line 221. 

Reviewer 3 Report

The manuscript presenteds the results of research on the influence of gender on time-domain and frequency domain heart rate measures during boring/uninterrupted 60-min driving.

I have two general remarks:

1. Some assumptions made while defining mental state raise my doubts. On what basis were the limits of 35 and 40 points (out of 65 maximum) adopted for the distribution of alert and fatigue segments (lines 126-127)? There is no references in the text. If it's authors assumption it must be justified. This division (assumed values) is crucial for the results of the research. 

2. It is very risky to base your entire conclusions on a single statistical test. As the authors themselves indicate in Tables 6 and 7, similar comparisons were made using other tests, for example the Paired t-test. It is worth running an alternative test and comparing the results for both tests. The authors themselves mention that the results they obtained on some issues are not consistent with the literature. One statistical test is not enough to confirm the validity of one's own results, which differ from others. I recommend performing alternative tests for the pairs under consideration and assessing the statistical significance of the differences between the values. This will definitely increase the scientific value of the work.

Detailed remarks:

  1. The test were performed between 9.30 and 17.30. This is long time and the perception and worsening of fatigue changes as the day goes on. The "limitations" should include information about the possible (unknown) influence of the time of day (morning, afternoon) on the test results.
  2. It has not been explained whether the points assigned to the segment by each of the 5 experts add up. I can guess this from the number of points taken for division. However, it should be clearly stated in the text.
  3. It is not clear what has been done (detailed procedure) to obtain HRV time series. The way to get RR intervals series is clear. However, it is not clear why "the HRV signal were derived from the RR intervals series using cubic spline interpolationand then converted into HRV time series by a sampling process", instead of being directly split into time series according to R peaks. Please explain.
  4. In place where symbols are used first time (lines 60-64) a reference to table 2 should be placed. 

Some small mistakes:

line 112 "Then, the subject was helped into the car" - reformulate

Table 5 4th row, 2nd column from the right - there is an artifact

Reviewer 4 Report

I found the style and writing of the manuscript adequate, and generally quite well-structured.

The introduction and methodology are adequate.

The discussion section should be used to contrast the results with other studies, not to introduce the results of your own research. I recommend re-structuring this part of the manuscript, introducing these data in the results section. In the discussion, I suggest including references such as Alonso, F., Esteban, C., Useche, S., & López de Cózar, E. (2016). Prevalence of Physical and Mental Fatigue Symptoms on Spanish Drivers and Its Incidence on Driving Safety. Advances in Psychology and Neuroscience, 1(2), 10-18. doi: 10.11648/j.apn.20160102.12; o Useche, S., Gómez, V., & Cendales, B. (2017). Stress-related psychosocial factors at work, fatigue, and risky driving behavior in Bus Rapid Transport (BRT) drivers. Accident Analysis & Prevention, 104, 106-114. doi: 10.1016/j.aap.2017.04.023.

In the conclusions you could add the limitations of the study, such as the small size of the sample, which limits the representativeness of your results, as well as the practical implication of this.

Round 2

Reviewer 1 Report

The authors provided responses to the reviewer's comments.  However, there are several new concerns/questions on the revised manuscript that need to be resolved.

  1. The results presented in the abstract is completely changed.  For instance, in the original manuscript 20 measures was statistically significant.  In the revised version, only 10 measures are statistically significant.  What is the reason behind this? There is no inclusion of new subjects in the experiments.  This raises a serious concern.
  2. Tables 4 and 5 present all p-values by their values.  The similar must be done for Tables 6 and 7.

Reviewer 3 Report

Thank you for your comprehensive answers to my questions and concerns. The newer version is much better than the previous one. Thanks to them, the results presented in the manuscript have become much more readable and understandable. Unfortunately, neither responce for remark 1 nor for remark 5 have been included in the new text version. This should be completed.
